# Addressing Imbalance for Class Incremental Learning in Medical Image Classification

## ABSTRACT

Deep convolutional neural networks have made significant breakthroughs in medical image classification, under the assumption that training samples from all classes are simultaneously available. However, in real-world medical scenarios, there's a common need to continuously learn about new diseases, leading to the emerging field of class incremental learning (CIL) in the medical domain. Typically, CIL suffers from catastrophic forgetting when trained on new classes. This phenomenon is mainly caused by the imbalance between old and new classes, and it becomes even more challenging with imbalanced medical datasets. In this work, we introduce two simple yet effective plug-in methods to mitigate the adverse effects of the imbalance. First, we propose a CIL-balanced classification loss to mitigate the classifier bias toward majority classes via logit adjustment. Second, we propose a distribution margin loss that not only alleviates the inter-class overlap in embedding space but also enforces the intra-class compactness. We evaluate the effectiveness of our method with extensive experiments on three benchmark datasets (CCH5000, HAM10000, and EyePACS). The results demonstrate that our approach outperforms state-of-the-art methods.

## CCS CONCEPTS

• **Computing methodologies** → **Computer vision**.

## KEYWORDS

Class incremental learning, medical image classification, class imbalance

## 1 INTRODUCTION

Nowadays, deep learning has emerged as a powerful tool across various fields, including the medical domain [2, 29, 32]. However, traditional deep learning methods often make assumptions about stationary and independent data distributions, which may be impractical in real-world scenarios. Most trained diagnosis models would be fixed once developed, while in real clinical practice, the distribution of medical data frequently undergoes shifts over time, primarily due to the continuous emergence of new diseases, treatment protocols, and patient data [6, 37]. Under such circumstances, the model needs to incorporate new class knowledge incrementally

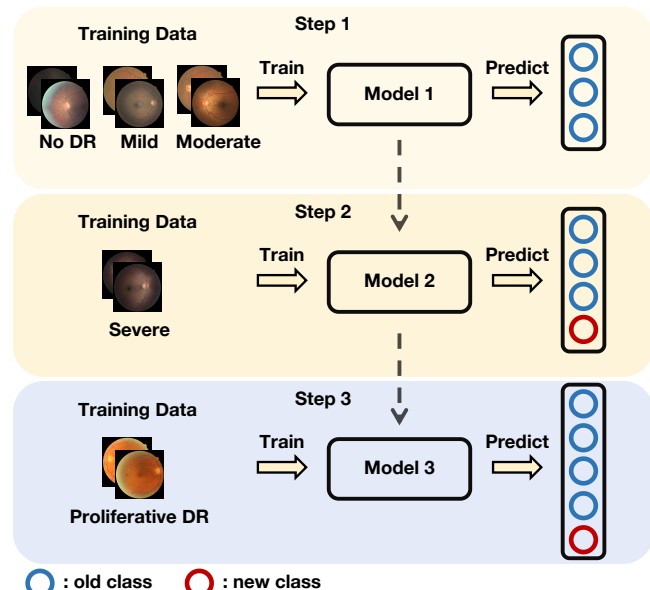

**Figure 1: Overview of the class incremental learning setting in medical image classification. During the incremental process, the training data is only provided for the current classes, while the data from previous steps is not accessible. At each step, the model is required to perform classification for all the classes seen so far.**

instead of retraining the model with all data available [24]. Therefore, in this work, we focus on class incremental learning in the medical domain.

Fig. 1 illustrates the setting of class incremental learning in medical Image classification. Taking the EyePACS dataset [7] as an example, the model is initially trained to classify three classes (*i.e.*, No DR, Mild, and Moderate). Subsequently, incremental classes (*e.g.*, Severe, and Proliferative DR) arrive in sequential steps to update the model. The classes introduced in different steps are disjoint, and the model must be able to predict all classes seen over time. However, when updating the model with only new classes, new data tends to erase previous knowledge. This phenomenon is known as catastrophic forgetting [15, 22].

For class incremental learning, imbalanced data between old and new classes is one of the primary reasons for catastrophic forgetting [23, 48]. To this end, numerous approaches have been proposed to store a small proportion of previous training data in memory and rehearse them when learning new classes [1, 18, 45]. However, the limited size of memory can also lead to an imbalance between old and new classes [28, 34]. Under this circumstance, the class imbalance will lead to (i) a classifier biased towards the new and majority classes; and (ii) the embeddings of new classes inevitably overlap

with the old ones in the feature space (i.e. the ambiguities problem). In addition to the class incremental learning imbalance, many real-life medical datasets exhibit significant class imbalance [32], with some classes having notably higher instances in training samples than others, *e.g.*, HAM10000 [38], and EyePACS [7], which further aggravate the catastrophic forgetting. Therefore, addressing data imbalance is crucial for class incremental learning in medical image classification.

In this paper, we propose two simple yet effective plug-in loss objectives to tackle two challenges caused by imbalance in class incremental learning. First, we propose a CIL-balanced classification loss instead of the traditional cross-entropy (CE) loss to avoid the issue of classifier bias. Specifically, we first adjust the logits based on the category frequency to place more emphasis on rare classes and then introduce a scale factor to further achieve a balance between old and new classes. Second, to alleviate the overlap of classes in the feature space, we propose a distribution margin loss, a novel improved margin loss, which not only facilitates to push away the distributions of old and new classes but also obtains the compact intra-class clustering. Extensive experiments on benchmark datasets under various settings verify the superiority of our method.

To summarize, the main contributions of this paper are:

- To reduce the classifier bias towards new and majority classes, we propose a CIL-balanced classification loss that emphasizes rare ones via logit adjustment.
- We introduce a novel distribution margin loss that can effectively separate the distributions of old and new classes to avoid ambiguities and realize the optimization of the intra-class compactness.
- Extensive experiments demonstrate that our method can effectively address the issue of data imbalance with the state-of-the-art performance achieved on three benchmark datasets: CCH5000, HAM10000, and EyePACS.

## 2 RELATED WORK

### 2.1 Class Incremental Learning

Class incremental learning aims to train a model from a sequence of classes, ensuring its performance across all the classes. Existing class incremental learning methods can be roughly divided into three groups: regularization-based, structure-based, and memory-based.

Regularization-based methods [10, 13, 18, 36] apply additional constraints to prevent the existing model from forgetting previous knowledge. LUCIR [18] constrains the orientation of the features to preserve the geometric configuration of old classes. PODNet [13] introduces a novel spatial distillation not only for the outputs of the final layer but also for the intermediate features to mitigate representation forgetting. However, regularization-based methods still suffer from feature degradation of old knowledge due to the limited access to old data [46].

Structure-based methods [14, 19, 27, 40, 46] aim to preserve the learned parameters associated with old classes while incrementally creating modules to enhance the model's capacity to acquire new knowledge. Recently, DER [46] adds a new feature extractor at each step and then concatenates the extracted features for

classification. DyTox [14] applies transformer [11] to incremental learning and dynamically expands task tokens when learning new classes. Nevertheless, dynamically adding new modules will lead to an explosion in the number of parameters and an increase in the independence between each feature extractor to harm performance in new classes [40].

Memory-based methods [3, 4, 34, 42, 45] address the challenge of forgetting by storing a limited number of representative samples from old classes in a memory buffer. iCaRL [34] learns the exemplar-based data representation and makes predictions using a nearest-mean-of-exemplars classifier. GEM [4] uses exemplars for gradient projection to overcome forgetting. Additionally, some approaches employ generative models to synthesize old class samples for data rehearsal [31, 35, 41] while other works consider saving feature embeddings instead of raw images [20]. In our work, we follow the memory-based approach to directly store a small subset of old class data for rehearsal.

### 2.2 Class Imbalance

Class imbalance is a key challenge for class incremental learning [18]. Due to the only access to the classes of the current step, the classifier is severely biased, and there is an inevitable overlap and confusion between the feature space representations of old and new classes [23]. Even with the limited size of the memory buffer, the biased optimization by imbalanced data between old and new classes is still a crucial problem that causes catastrophic forgetting [28, 34]. To cope with it, SS-IL [1] isolates the computation of the softmax probabilities on old and new classes for bias compensation. BiC [45] introduces a bias correction layer to address the bias in the last fully connected layer.

In real-world medical scenarios, most existing datasets contain highly imbalanced numbers of samples [32], which leads to a more severe forgetting. To the best of our knowledge, LO2LN [6] is the first attempt to address the problem of class incremental learning in medical image classification. First, they utilize the class-balanced focal loss [8] to avoid the classifier bias. However, the class-balanced focal loss is not specialized and efficient for incremental learning. Second, they introduce the margin ranking loss [18] to separate old and new classes. We argue that this constraint may not be sufficiently robust, resulting in large clusters within classes (intra-class) and potential overlaps between classes (inter-class). By contrast, in this paper, we propose two simple yet effective plug-in loss objectives: (i) a CIL-balanced classification loss to alleviate prediction bias by adjusting the logits, and (ii) a distribution margin loss that can push the distributions of old and new classes away and provide more compact intra-class clustering simultaneously.

## 3 METHOD

In this section, we first outline the setting of class incremental learning in medical image classification (Sec. 3.1). Then, we provide a detailed description of the two proposed loss objectives: CIL-balanced classification loss (Sec. 3.2) and distribution margin loss (Sec. 3.3).

## 3.1 Setting and Notation

Class incremental learning aims to train a model from a sequence of data incrementally. Specifically, we denote the sequence of tasks as $\mathcal{D} = \{\mathcal{D}_1, \mathcal{D}_2, \mathcal{D}_3, ..., \mathcal{D}_T\}$, where $\mathcal{D}_t = (X_t, \mathcal{Y}_t) = \{(x_i^t, y_i^t)\}_{i=1}^{n_t}$ represents the training set from step $t$ with $n_t$ instances. Here, $x_i^t \in X_t$ is a sample and $y_i^t \in \mathcal{Y}_t$ is the corresponding label. The label space of the model is all seen classes $\mathcal{Y}_{i:t} = \bigcup_{i=1}^{t} \mathcal{Y}_i$, where $\mathcal{Y}_t \cap \mathcal{Y}_{t'} = \emptyset$ for all $t \neq t'$. Inspired by memory-based methods [3, 34, 45], our method consistently samples $m$ representative instances from each old class and store them in a memory buffer $\mathcal{M}_t$, which is updated after the training step $t$ is completed. It should be mentioned that only data from $\hat{\mathcal{D}}_t = \mathcal{D}_t \cup \mathcal{M}_{t-1}$ is available for training during the $t$-th step.

Classically, the model at step t can be written as the composition of two functions: $f^t = f_\theta^t \circ f_\phi^t(\cdot)$, where $f_\phi^t$ represents a feature extractor, and $f_\theta^t$ represents a classifier. For an input sample $x_i$, its feature representation is denoted as $h_i^t = f_\phi^t(x_i)$. We employ cosine normalization [18] as the classifier $f_\theta^t$. Consequently, the predicted logit $p_{i,c}^t$ for class $c$ at step $t$ can be calculated from $h_i^t$ as:

$$p_{i,c}^t = \eta \left\langle h_i^t, w_c \right\rangle, \tag{1}$$

where $w_c$ are the weights for class $c$ in the classifier layer, $\eta$ is a learnable scalar, and $\langle \cdot, \cdot \rangle$ denotes the cosine similarity between two vectors.

## 3.2 CIL-Balanced Classification Loss

As claimed in previous works [28, 32], the inherent imbalance in medical datasets and the imbalance in class incremental learning can lead to a biased classifier. Inspired by [30], we aim to mitigate this issue by adjusting the logits according to category frequency. However, for a memory-based method in class incremental learning, only the data from $\hat{\mathcal{D}}_t$ is available at step $t$, which consists of the memory buffer $\mathcal{M}_{t-1}$ and the training set $\mathcal{D}_t$. Hence, we define the category frequency as follows:

$$r_c = \begin{cases} \frac{m}{\left|\hat{\mathcal{D}}_t\right|}, & \text{if } c \in \mathcal{Y}_{1:t-1}, \\ \frac{q_c}{\left|\hat{\mathcal{D}}_t\right|}, & \text{if } c \in \mathcal{Y}_t, \end{cases} \tag{2}$$

where $q_c$ is the number of training samples for class $c$, and $|\cdot|$ is the cardinality of a given set. After that, we add $log\, r_c$ to the output logits during training. Thus, the logit-balanced classification loss can be formulated as:

$$\mathcal{L}_{lbc} = -\frac{1}{\left|\hat{\mathcal{D}}_t\right|} \sum_{i \in \hat{\mathcal{D}}_t} log \frac{exp\left(p_{i,y_i}^t + log\, r_{y_i}\right)}{\sum_{j \in \mathcal{Y}_{1:t}} exp\left(p_{i,j}^t + log\, r_j\right)}. \tag{3}$$

To explain how our method works, we reformulate Eq. 3 into Eq. 4 by introducing $v_{y_i, j} := log\, r_j - log\, r_{y_i}$, which are defined as follows:

$$\mathcal{L}_{lbc} = -\frac{1}{\left|\hat{\mathcal{D}}_t\right|} \sum_{i \in \hat{\mathcal{D}}_t} log \frac{exp\left(p_{i,y_i}^t\right)}{\sum_{j \in \mathcal{Y}_{1:t}} exp\left(p_{i,j}^t + v_{y_i, j}\right)}, \tag{4}$$

where:

$$v_{y_i, j} = \begin{cases} log\, \frac{q_j}{m} \;\; [> 0], & \text{if } y_i \in \mathcal{Y}_{1:t-1}, j \in \mathcal{Y}_t, \\ log\, \frac{m}{m} \;\; [= 0], & \text{if } y_i \in \mathcal{Y}_{1:t-1}, j \in \mathcal{Y}_{1:t-1}, \\ log\, \frac{m}{q_{y_i}} \;\; [< 0], & \text{if } y_i \in \mathcal{Y}_t, j \in \mathcal{Y}_{1:t-1}, \\ log\, \frac{q_j}{q_{y_i}}, & \text{if } y_i \in \mathcal{Y}_t, j \in \mathcal{Y}_t. \end{cases} \tag{5}$$

It is known that traditional softmax loss necessitates $p_{i,y_i}^t > p_{i,j}^t$ for the accurate classification of sample $x_i$. In order to prioritize the learning of old and rare classes, we employ the following logit adjustment strategy. Specifically, when $y_i \in \mathcal{Y}_{1:t-1}$ and $j \in \mathcal{Y}_t$ (the first line in Eq. 5), we instead require $p_{i,y_i}^t > p_{i,j}^t + \log\left(q_j/m\right)\,[> 0]$. Hence, it is clear that we require a larger $p_{i,y_i}^t$, which makes the training process place more emphasis on old class $y_i$ than previously. However, if both $y_i$ and $j$ are within $\mathcal{Y}_{1:t-1}$ (the second line in Eq. 5), the logit remains unchanged, since they are both old classes with the same memory size.

For the other two cases when $y_i \in \mathcal{Y}_t$. If $j \in \mathcal{Y}_{1:t-1}$ (the third line in Eq. 5), the term $\log\left(m/q_{y_i}\right) < 0$ suggests that old class $j$ receives more emphasis. If $j \in \mathcal{Y}_t$ (the fourth line in Eq. 5), more emphasis is placed on the class $y_i$ when it has fewer instances, and conversely, the focus is on class $j$ when the size $q_j$ is smaller. Therefore, the logit-balanced classification loss can effectively reduce the bias towards new and frequent classes.

To further control the balance between the old and new classes, we introduce a scale factor $\gamma$:

$$\gamma_c = \begin{cases} \alpha, & \text{if } c \in \mathcal{Y}_{1:t-1}, \\ 1, & \text{if } c \in \mathcal{Y}_t, \end{cases} \tag{6}$$

where $\alpha \in [0, 1]$ is a trade-off coefficient for each dataset. With the help of this scale factor, the CIL-balanced classification loss can be written as:

$$\mathcal{L}_{cbc} = -\frac{1}{\left|\hat{\mathcal{D}}_t\right|} \sum_{i \in \hat{\mathcal{D}}_t} log \frac{\gamma_{y_i} \cdot exp\left(p_{i,y_i}^t\right)}{\sum_{j \in \mathcal{Y}_{1:t}} \gamma_j \cdot exp\left(p_{i,j}^t + v_{y_i, j}\right)}, \tag{7}$$

which reduces the output values for the old classes while maintaining the outputs for the new classes unchanged, thereby encouraging the model to produce larger logits for these old ones. Consequently, this scaling strategy further mitigates the issue of imbalance in class incremental learning. In this context, although a decrease in $\alpha$ improves the significance of old classes, it may affect the model's learning of new ones. Thus, determining the optimal $\alpha$ becomes crucial for achieving a better trade-off (see Sec. 4.4). Notably, when $\alpha$ is assigned a value of 1, the current CIL-balanced classification loss degrades to the logit-balanced classification loss (Eq. 4).

## 3.3 Distribution Margin Loss

In class incremental learning, the representations of the old and new classes would be easily overlapped in the deep feature space [47]. To address this issue, margin loss [5] is introduced to avoid the ambiguities between old and new classes. In detail, the vanilla margin loss aims to ensure that the distance from the anchor to the positive (embedding of the ground-truth old class) is less than the distance of the anchor from the negative (embedding of the new

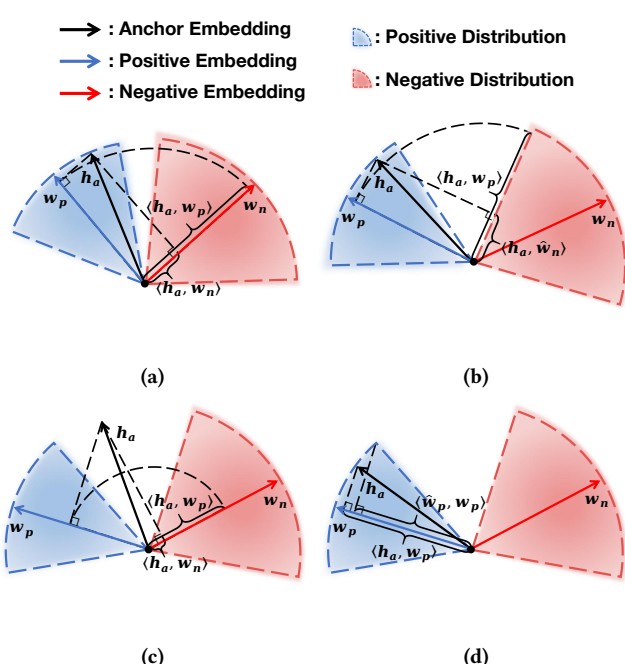

**: Anchor Embedding**
**: Positive Embedding**
**: Negative Embedding**
**: Positive Distribution**
**: Negative Distribution**

(a)          (b)

(c)          (d)

**Figure 2: (a) The vanilla margin loss forces the cosine similarity between $h_a$ and $w_p$ to be larger than that between $h_a$ and $w_n$ without considering the distribution separation. (b) Our distribution margin loss aims to push $h_a$ away from the distribution of the negative class instead of just $w_n$, thus mitigating feature space overlap. (c) The vanilla margin loss fails to minimize the intra-class distance adequately, which may result in $h_a$ being distant from the center of its ground-truth class. (d) The distribution margin loss ensures that $h_a$ remains within its corresponding class distribution, enhancing intra-class compactness.**

class) to meet a predefined margin $m$, which can be computed as:

$$\mathcal{L}_m = \sum_{i \in \mathcal{M}_{t-1}} \sum_{c \in \mathcal{Y}_t} max \left\{ 0, \langle h_i^t, w_c \rangle - \langle h_i^t, w_{y_i} \rangle + m \right\}, \quad (8)$$

where $\langle \cdot, \cdot \rangle$ denotes the cosine similarity and the margin $m$ is set to 0.4 for all experiments.

However, the vanilla margin loss exhibits two limitations. First, it only focuses on the triplet: anchor, positive, and negative embeddings. Even if the distance from the anchor to the negative exceeds that to the positive by a margin $m$, their distributions may remain close or even overlap, thereby introducing potential ambiguities in classification (shown in Fig. 2a). Second, while the vanilla margin loss aims to separate the ground-truth old class from new classes (maximizing inter-class distance), it fails to adequately address the minimization of intra-class distance, often leading to large intra-class clustering (shown in Fig. 2c).

To address the above limitations, we try to restore the class distribution and design a novel distribution margin loss that contains two loss terms. The first term optimizes the triples by ensuring that the distance from the anchor to the positive embedding is less than its distance to the negative class distributions by the margin

**Input**: Incremental task data $\mathcal{D}_t$, Memory exemplars: $\mathcal{M}_{t-1}$
**Output**: Updated current model
1: // Training process in incremental steps ($t \geq 2$)
2: $\hat{\mathcal{D}}_t = \mathcal{D}_t \cup \mathcal{M}_{t-1}$;                          ▷ Rehearsal
3: **repeat**
4:     $\mathcal{L}_{cbc} \leftarrow$ Eq. 7;          ▷ CIL-Balanced Classification Loss
5:     $\mathcal{L}_{dm} \leftarrow$ Eq. 9;               ▷ Distribution Margin Loss
6:     $\mathcal{L}_{kd} \leftarrow$ Eq. 11;              ▷ Knowledge Distillation Loss
7:     // Update the current model via optimizing $\mathcal{L}_{all}$
8:     $\mathcal{L}_{all} \leftarrow$ Eq. 12;                          ▷ Overall Loss
9: **until** reaches predefined epoch

$m$, rather than merely to the negative embeddings. By optimizing this term, the distribution margin loss can push the samples of old classes away from the new class distributions to facilitate the inter-class separation (shown in Fig. 2b). The second term attempts to maintain the anchor embedding within the distribution range of its corresponding class, thus obtaining compact intra-class clustering (shown in Fig. 2d). Accordingly, the distribution margin loss can be formulated as:

$$\mathcal{L}_{dm} = \sum_{i \in \mathcal{M}_{t-1}} \sum_{c \in \mathcal{Y}_t} max \left\{ 0, \langle h_i^t, \hat{w}_c \rangle - \langle h_i^t, w_{y_i} \rangle + m \right\}$$
$$+ \sum_{i \in \mathcal{M}_{t-1}} max \left\{ 0, \langle \hat{w}_{y_i}, w_{y_i} \rangle - \langle h_i^t, w_{y_i} \rangle \right\}, \quad (9)$$

where $\hat{w}_c$ represents the distribution range of class $c$. Specifically, we model the data distribution of each class in the feature space by applying a Gaussian distribution around their centroids. However, due to the imbalanced number of samples across different classes, the features of classes with limited instances may get squeezed into a narrow area in the feature space [43]. As a result, we assign a larger distribution range to the majority classes and a more restricted range to the minority classes:

$$\hat{w}_c = w_c + \eta * \hat{r}_c, \quad \hat{r}_c = \frac{q_c}{\sum_{i \in \mathcal{Y}_{1:t}} q_i}, \quad (10)$$

where $\hat{r}_c$ represents the inherent ratio of class $c$ among all seen classes, and $\eta \sim \mathcal{N}(0, 1)$ is a Gaussian noise which has the same dimension as the classifier weight.

To prevent forgetting and maintain the discrimination ability, we also apply knowledge distillation loss [17] to build a mapping between the old and the current model:

$$\mathcal{L}_{kd} = \frac{1}{|\hat{\mathcal{D}}_t|} \sum_{i \in \hat{\mathcal{D}}_t} \sum_{c \in \mathcal{Y}_{1:t-1}} \left\| p_{i,c}^t - p_{i,c}^{t-1} \right\|. \quad (11)$$

Therefore, the overall loss is defined as:

$$\mathcal{L}_{all} = \mathcal{L}_{cbc} + \lambda_d \mathcal{L}_{dm} + \lambda_k \mathcal{L}_{kd}, \quad (12)$$

where $\lambda_d$ and $\lambda_k$ are the hyper-parameters for balancing the importance of each loss. We show the guideline of our method at incremental step $t$ in Alg. 1.

# 4 EXPERIMENTS

## 4.1 Experimental Setups

**Datasets.** Following the benchmark setting [6], we evaluate the performance on CCH5000 [21], HAM10000 [38], and EyePACS [7].

- **CCH5000**: consists of histological images in human colorectal cancer. This dataset contains 8 different classes with 625 images per class: tumor, stroma, complex, lympho, debris, mucosa, adipose, and empty.
- **HAM10000**: consists of 10,015 skin cancer images, including seven types of skin lesions: melanoma, melanocytic nevus, basal cell carcinoma, actinic keratosis, benign keratosis, dermatofibroma, and vascular lesions. The distribution ratios for each type are as follows: 3.27%, 5.13%, 10.97%, 1.15%, 11.11%, 66.95%, and 1.42%, which indicates a severe class imbalance.
- **EyePACS**: is commonly used for the task of diabetic retinopathy (DR) classification. EyePACS dataset contains 35,126 retina images for training, which are categorized into five stages of DR. Specifically, there are 25,810 images labeled as no DR, 2,443 as mild DR, 5,292 as moderate DR, 873 as severe DR, and 708 as proliferative DR images. It is worth noting that this dataset is also highly imbalanced.

**Evaluation protocols.** Following the experimental protocols in [6], we evaluate our method for different scenarios, such as 4-1, 4-2, 3-1, and 3-2. In each scenario, the numbers indicate the number of base and new classes, respectively. For example, considering the HAM10000 dataset with 7 classes, the scenario of 3-2 corresponds to learning 3 classes at the initial step and subsequently adding 2 new classes at each incremental step, requiring a total of 3 training steps.

**Metrics.** Following previous work [6], we evaluate our method based on two standard metrics: Average Accuracy ($Acc$) and Average Forgetting ($Fgt$). Let $a_{t,i}$ be the accuracy of the model evaluated on the test set of classes in $\mathcal{Y}_i$ after training on the first $t$ steps. The Average Accuracy is defined as:

$$Acc = \frac{1}{T}\sum_{t=1}^{T}\left(\frac{1}{t}\sum_{i=1}^{t} a_{t,i}\right), \tag{13}$$

which measures the average classification accuracy of the model until step $T$. The Average Forgetting is defined as:

$$Fgt = \frac{1}{T}\sum_{t=1}^{T}\left[\frac{1}{t-1}\sum_{i=1}^{t-1}\max_{j\in[1,t-1]}\left(a_{j,i} - a_{t,i}\right)\right], \tag{14}$$

which measures an estimate of how much the model forgets by averaging the decline in accuracy from the peak performance to its current performance.

**Compared methods.** To demonstrate the superiority of our method, we first compare it to classical incremental learning approaches: iCaRL [34], UCIR [18], PODNet [13], and DER [46]. Besides, we also compare to the current state-of-the-art method: LO2LN [6].

**Implementation details.** As in [6], we adopt a cosine normalization classifier with a ResNet-18 [16] backbone pre-trained on the ImageNet [9]. Our method is implemented in PyTorch [33], and we employ SGD with a momentum value of 0.9 and weight decay of 0.0005 for optimization. During training, the batch size is set to 32 for the CCH5000 and HAM10000 datasets and 128 for the EyePACS dataset in each learning step. Note that, for a fair comparison, we

| Method | 4-2 (3 steps) | | 4-1 (5 steps) | |
|---|---|---|---|---|
| | *Acc* | *Fgt* | *Acc* | *Fgt* |
| iCaRL [34] | 93.0±0.2 | 6.8±1.0 | 91.1±1.8 | 9.0±3.3 |
| UCIR [18] | 93.9±0.3 | 4.4±0.9 | 92.0±1.0 | 5.5±2.6 |
| PODNET [13] | 92.0±0.3 | 5.2±0.4 | 89.2±0.5 | 6.0±1.2 |
| DER [46] | 93.0±0.5 | 6.4±1.4 | 91.0±1.7 | 5.6±1.9 |
| LO2LN [6] | 94.6±0.4 | 4.0±0.8 | 94.5±0.8 | 3.9±2.0 |
| **Ours** | **95.5±0.2** | **2.3±0.7** | **95.2±0.2** | **2.1±1.5** |

Table 1: Experimental results on CCH5000 under three different class orders. Numbers in bold denote the best results.

| Method | 3-2 (3 steps) | | 3-1 (5 steps) | |
|---|---|---|---|---|
| | *Acc* | *Fgt* | *Acc* | *Fgt* |
| iCaRL [34] | 76.3±3.1 | 20.1±12.6 | 68.3±2.8 | 25.3±4.5 |
| UCIR [18] | 79.1±1.4 | 16.8±9.5 | 74.1±3.1 | 16.3±9.1 |
| PODNET [13] | 75.6±2.2 | 20.5±2.1 | 66.3±2.3 | 17.3±4.8 |
| DER [46] | 76.2±2.8 | 24.8±10.9 | 66.9±4.5 | 24.7±4.8 |
| LO2LN [6] | 82.0±1.3 | 12.8±3.3 | 78.1±3.4 | 10.1±3.9 |
| **Ours** | **85.0±3.1** | **8.0±3.5** | **80.9±2.9** | **5.2±2.5** |

Table 2: Experimental results on HAM10000 under three different class orders. Numbers in bold denote the best results.

| Method | 3-1 (3 steps) | |
|---|---|---|
| | *Acc* | *Fgt* |
| iCaRL [34] | 64.4±3.3 | 17.8±4.6 |
| UCIR [18] | 70.2±7.6 | 15.4±11.4 |
| PODNET [13] | 63.3±5.4 | 22.8±4.9 |
| DER [46] | 58.7±9.4 | 30.2±6.9 |
| LO2LN [6] | 81.9±2.5 | -0.2±0.8 |
| **Ours** | **82.8±2.8** | **-0.5±0.7** |

Table 3: Experimental results on EyePACS under three different class orders. Numbers in bold denote the best results.

use the same memory setting for every compared method, *i.e.,* a fixed number of 20 training examples per class are selected via the herding selection strategy [44] and stored in memory $\mathcal{M}$. Furthermore, we conduct all experiments on three different class orders and report the means ± standard deviations over three runs.

## 4.2 Performance Comparison

As shown in Tab. 1, 2, and 3, we report the experimental results on three benchmark datasets: CCH5000, HAM10000, and EyePACS, respectively.

**CCH5000.** We can see that our method achieves state-of-the-art performance in terms of $Acc$ and $Fgt$ on both settings. Specifically, our method surpasses LO2LN by 1.7% on the 4-2 setting and 1.8% on the 4-1 setting in terms of $Fgt$, indicating the effectiveness of our method in overcoming forgetting.

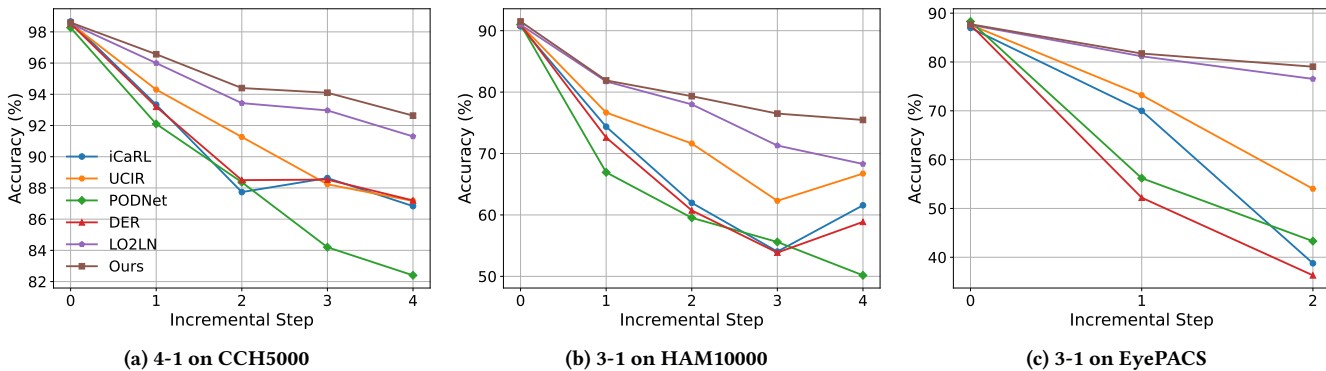

(a) 4-1 on CCH5000      (b) 3-1 on HAM10000      (c) 3-1 on EyePACS

Figure 3: Accuracy at each step on CCH5000, HAM10000, and EyePACS.

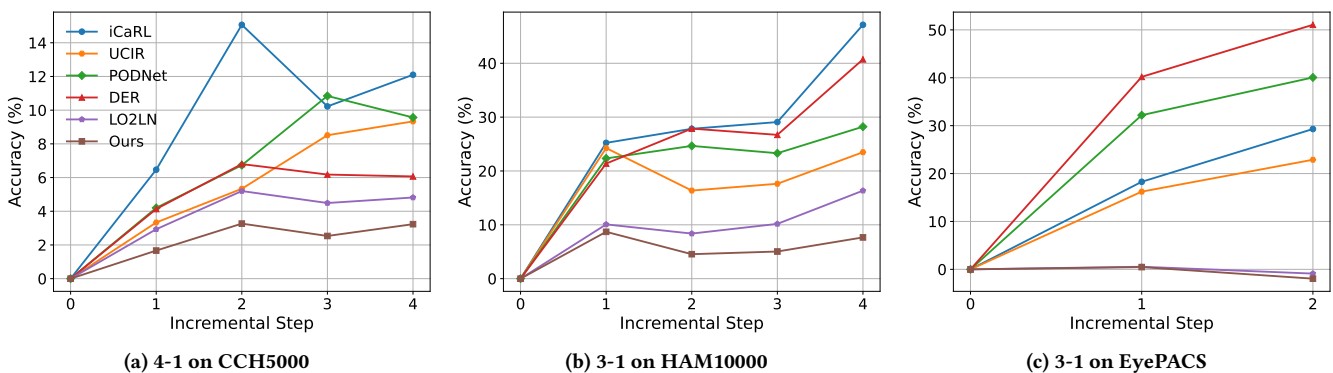

(a) 4-1 on CCH5000      (b) 3-1 on HAM10000      (c) 3-1 on EyePACS

Figure 4: Forgetting at each step on CCH5000, HAM10000, and EyePACS.

**HAM10000.** Different from the CCH5000 dataset, the HAM10000 dataset is a highly imbalanced dermoscopy dataset. Experimental results demonstrate that our method significantly improves the performance on the HAM10000 dataset, benefiting from the strong ability to address class imbalance. To be more specific, compared with the SOTA method, we improve the accuracy from 82.0% to 85.0% on the 3-2 setting. On the 3-1 setting, we achieve an overall performance of 80.9%, which is 2.8% higher than LO2LN's 78.1%. Moreover, the average forgetting is also reduced by 4.8% (3-2 setting) and 4.9% (3-1 setting).

**EyePACS.** Furthermore, we present a comparison of different methods on the challenging EyePACS dataset. Our proposed method not only demonstrates significantly higher average accuracy but also achieves lower average forgetting than the other baselines. Notably, it surpasses LO2LN by 0.9% in terms of $Acc$ and outperforms PODNet and DER by substantial margins of 19.5% and 24.1%, respectively.

## 4.3 Analysis of Incremental Performance

**Accuracy.** As shown in Fig. 3, we display the average incremental performance of each step for three datasets. According to these curves, it is evident that the performances of all methods are similar in the first step, but the baselines suffer from a significant drop as the learning steps increase. In contrast, our method effectively slows down the drop, leading to an increasing gap between the baselines

| $\mathcal{L}_{cbc}$ | $\mathcal{L}_{dm}$ | $Acc$ | $Fgt$ |
|:---:|:---:|:---:|:---:|
| ✗ | ✗ | 67.6±1.6 | 30.4±11.9 |
| ✗ | ✓ | 80.2±2.8 | 8.6±4.0 |
| ✓ | ✗ | 83.6±2.9 | 13.1±6.0 |
| ✓ | ✓ | **85.0±3.1** | **8.0±3.5** |

**Table 4: Performance contribution of each component on the HAM10000 3-2 setting.**

and our method over time. This demonstrates that our method benefits class incremental learning in medical image classification and outperforms prior works.

**Forgetting.** Fig. 4 depicts the average forgetting across each incremental step for three datasets. The forgetting of most methods increases rapidly as new classes arrive, while our method consistently outperforms the SOTA methods, indicating improved resilience to catastrophic forgetting.

## 4.4 Ablation Study

**Impact of each component.** In Tab. 4, we present an ablation analysis on the HAM10000 3-2 setting to evaluate the effect of each proposed component. The first row refers to the baseline, which is

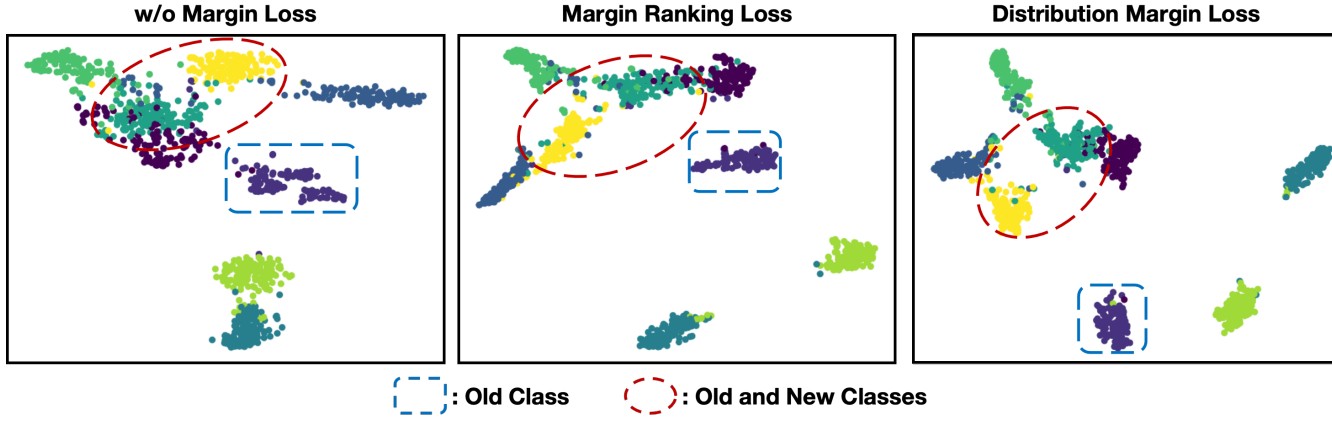

**Figure 5: The t-SNE visualization of feature distributions of w/o margin loss (left), margin ranking loss (middle), and our distribution margin loss (right) on the CCH5000 dataset.**

| Classification loss | Acc | Fgt |
|---|---|---|
| CE | 80.2±2.8 | 8.6±4.0 |
| Focal [25] | 81.5±2.4 | 12.6±5.2 |
| CB Focal [8] | 82.9±3.1 | 14.5±6.3 |
| **logit-balanced (Eq. 4)** | 83.1±3.0 | 14.2±6.0 |
| **CIL-balanced (Eq. 7)** | **85.0±3.1** | **8.0±3.5** |

**Table 5: Performance of different classification losses on the HAM10000 3-2 setting.**

| Margin loss | Acc | Fgt |
|---|---|---|
| w/o margin loss | 83.6±2.9 | 13.1±6.0 |
| Margin ranking loss [18] | 83.9±2.4 | 12.6±5.1 |
| **Distribution margin loss (Eq. 9)** | **85.0±3.1** | **8.0±3.5** |

**Table 6: Performance of different margin losses on the HAM10000 3-2 setting.**

| Method | Acc | | Fgt | |
|---|---|---|---|---|
| iCARL | 68.3±2.8 | | 25.3±4.5 | |
| + $\mathcal{L}_{dm}$ | 69.2±3.1 | +0.9 | 23.7±8.1 | +1.6 |
| + $\mathcal{L}_{cbc}$ | 70.8±2.8 | +2.5 | 21.9±6.5 | +3.4 |
| + $\mathcal{L}_{cbc}$ + $\mathcal{L}_{dm}$ | **71.6±2.4** | +3.3 | **19.6±7.3** | +5.7 |
| UCIR | 74.1±3.1 | | 16.3±9.1 | |
| + $\mathcal{L}_{dm}$ | 75.7±4.6 | +1.6 | 13.9±8.2 | +2.4 |
| + $\mathcal{L}_{cbc}$ | 76.4±3.4 | +2.3 | 2.1±4.0 | +14.2 |
| + $\mathcal{L}_{cbc}$ + $\mathcal{L}_{dm}$ | **77.1±4.6** | +3.0 | **1.5±6.0** | +14.8 |

**Table 7: Impact of integrating the CIL-balanced classification loss $\mathcal{L}_{cbc}$ and the distribution margin loss $\mathcal{L}_{dm}$ with existing methods on the HAM1000 3-1 setting. The red highlights the relative improvement.**

trained with the cross-entropy loss (CE) and the knowledge distillation loss $\mathcal{L}_{kd}$. Firstly, we observe that the distribution margin loss $\mathcal{L}_{dm}$ brings a significant contribution when applied alone, improving the performance by 12.6% in terms of *Acc*. Secondly, when we replace CE with the CIL-balanced classification loss $\mathcal{L}_{cbc}$, the average accuracy is improved from 67.6% to a notable 83.6%. Finally, the combination of $\mathcal{L}_{dm}$ and $\mathcal{L}_{cbc}$ further improves the performance, demonstrating the effect of both proposed components.

**Effect of CIL-Balanced Classification Loss.** We investigate the impact of different classification losses on the HAM10000 3-2 setting when combined with the knowledge distillation loss $\mathcal{L}_{kd}$ and our distribution margin loss $\mathcal{L}_{dm}$. As shown in Tab. 5, we present results of using cross-entropy loss (CE), focal loss (Focal) [25], class-balanced focal loss (CB Focal) [8], and our proposed methods (logit-balanced and CIL-balanced). It can be observed that both of our proposed methods consistently outperform the other classification loss objectives, indicating the effectiveness of them to address the imbalance issue. Furthermore, the CIL-balanced classification loss (Eq. 7) achieves an additional 1.9% improvement compared to the logit-balanced classification loss (Eq. 4), benefiting from the scale factor $\gamma$ to strengthen the learning of old classes.

**Effect of Distribution Margin Loss.** To verify the effectiveness of our distribution margin loss objectives, we conduct experiments on the HAM10000 3-2 setting combined with the knowledge distillation loss $\mathcal{L}_{kd}$ and our CIL-balanced classification loss $\mathcal{L}_{cbc}$. The results presented in Tab. 6 demonstrate that our distribution margin loss brings significant improvements compared to cases without the margin loss and with the margin ranking loss [18]. To further illustrate the advantages of our method, we present t-SNE [39] visualizations of feature distributions with different margin loss objectives for the CCH5000 dataset, as shown in Fig. 5. In the absence of margin loss, we observe large intra-class clusters (blue rectangle) and significant inter-class overlap in feature space (red circle). When employing the margin ranking loss, the issue of

ACM MM, 2024, Melbourne, Australia

Anonymous Authors

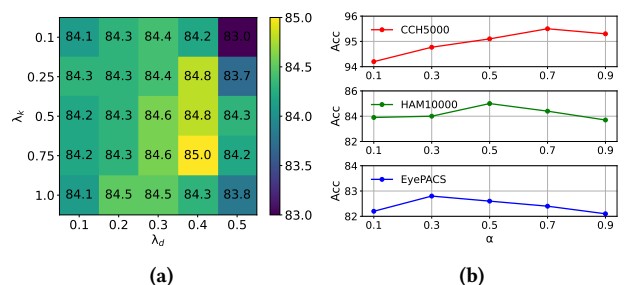

(a)                    (b)

Figure 6: Sensitivity study of hyper-parameters. (a) $\lambda_d$ and $\lambda_k$ on the HAM10000 3-2 setting. (b) $\alpha$ on three datasets.

overlap is mitigated to some extent (red circle) compared to the method without margin loss. Finally, by optimizing our distribution margin loss, we achieve a more pronounced separation between the distributions of old and new classes (red circle), while simultaneously ensuring that the representations of old classes become more compact (blue rectangle).

**Ability to integrate with other existing methods.** Our proposed methods can be easily integrated with other existing CIL methods. To demonstrate this, we conduct experiments utilizing iCaRL [34] and UCIR [18] on the HAM10000 3-1 setting. Specifically, we replace the classification loss in each baseline with our CIL-balanced classification loss and incorporate our distribution margin loss. As shown in Tab. 7, the accuracy (Acc) for both baselines can be improved by about 3% with the integration of our methods. More notably, our approaches effectively reduce forgetting (Fgt) by 5.7% and 14.8% for iCaRL and UCIR, respectively.

**Sensitivity study of hyper-parameters.** In this paper, there are three hyper-parameters during training: the weight of the distribution margin loss $\lambda_d$, the weight of the knowledge distillation loss $\lambda_k$, and the trade-off coefficient $\alpha$. We first conduct experiments to explore the impacts of different $\lambda_d$ and $\lambda_k$ on the HAM10000 3-2 setting. As shown in Fig. 6a, we vary $\lambda_d$ within the range of $\{0.1, 0.2, 0.3, 0.4, 0.5\}$, and $\lambda_k$ within the range of $\{0.1, 0.25, 0.5, 0.75, 1.0\}$, resulting in a total of 25 compared results. From the results, we consistently observe satisfactory performance from our model, demonstrating its robustness to the selection of $\lambda_d$ and $\lambda_k$.

To investigate the impact of different values of $\alpha$ on addressing the imbalance between the old and new classes, we evaluate the accuracy by varying $\alpha$ from $\{0.1, 0.3, 0.5, 0.7, 0.9\}$ on three benchmark datasets. As shown in Fig. 6b, the results indicate that the accuracy gradually improves as $\alpha$ grows larger initially, while it starts to decline when $\alpha$ is close to 1. Since the data distribution differs across datasets, the selection of the trade-off coefficient $\alpha$ also varies. Specifically, the optimal values of $\alpha$ for the three datasets are 0.7, 0.5, and 0.3, respectively.

**Longer incremental learning.** In class incremental learning, a key challenge is catastrophic forgetting, which becomes more pronounced as the number of learning classes increases [12, 18, 34]. To quantify the robustness of our method in overcoming catastrophic forgetting, we evaluate it on two longer-step protocols: 50-10 (6 steps) and 50-5 (11 steps), employing the more challenging CIFAR100 dataset. Following the experimental protocol outlined

| Method | 50-10 (6 steps) | | 50-5 (11 steps) | |
| --- | --- | --- | --- | --- |
| | Conv | LT | Conv | LT |
| UCIR [18] | 61.2 | 42.7 | 58.7 | 42.2 |
| PODNET [13] | 63.2 | 44.1 | 61.2 | 44.0 |
| LWS [26] | 64.6 | 44.4 | 62.6 | 44.4 |
| **Ours** | **65.8** | **48.2** | **63.9** | **47.5** |

Table 8: Average accuracy on CIFAR100 in the conventional (Conv) and long-tailed (LT) scenarios. Numbers in bold denote the best results.

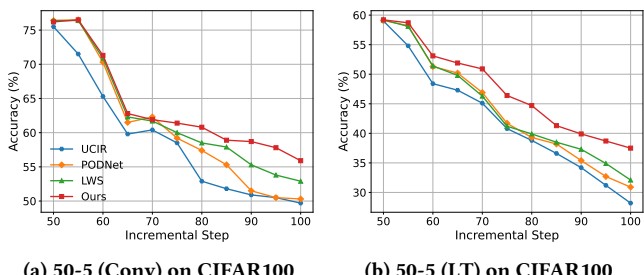

(a) 50-5 (Conv) on CIFAR100          (b) 50-5 (LT) on CIFAR100

Figure 7: Incremental accuracy on CIFAR100 50-5 setting for both conventional (Conv) and long-tailed (LT) scenarios.

in [26], we conduct experiments under both conventional (Conv) and long-tail (LT) scenarios. In the conventional scenario, each class has 500 training samples for training. Conversely, the long-tailed scenario follows an exponential decay in sample sizes across classes, where the ratio between the least and the most frequent class is 0.01. As illustrated in Tab. 8, our method achieves superior results in all settings. Specifically, we observe a more significant improvement in the long-tail scenario, further validating the effectiveness of our method in addressing the class imbalance problem in class incremental learning. Furthermore, we present the dynamic performance changes during the incremental learning process in Fig. 7. It is evident that with more learning steps, the gap between the baselines and our method widens, and our method's performance remains superior across different scenarios (conventional and long-tailed) throughout most of the learning steps.

## 5 CONCLUSION

In this paper, we propose two simple yet effective plug-in loss functions for class incremental learning in medical image classification. First, to address the challenge of classifier bias caused by class imbalance, we introduce a CIL-balanced classification loss via logit adjustment. Second, we propose a novel distribution margin loss that aims to enforce inter-class discrepancy and intra-class compactness simultaneously. Our extensive experimental evaluation demonstrates the state-of-the-art performance of our method across various scenarios on medical image datasets: CCH5000, HAM10000, and EyePACS.

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
