# OpenReview forum: "Addressing Imbalance for Class Incremental Learning in Medical Image Classification"
_acmmm.org/ACMMM/2024/Conference — MM2024 Poster_

### Official Review · Reviewer_jFBd · 2024-05-24

**Rating:** 4
**Confidence:** 2

**Summary:**

This paper addresses class imbalance in class incremental learning for medical image segmentation. It identifies the imbalance between old and new classes as a major concern and proposes two strategies to mitigate this issue: CIL-balanced classification loss and distribution marginal loss. Experimental results demonstrate the effectiveness of these strategies.

**Strengths:**

- The motivation is clear, and the proposed design seems reasonable.
- The experimental results, including comparisons with state-of-the-art methods and ablation studies, demonstrate the effectiveness.

**Limitations:**

- Please justify why \(\alpha\) is restricted to the range [0-1] in Row 321. In my understanding, this is a hyperparameter. Furthermore, the range of [0-1] for the old class seems too close to the weight for the new class, indicating a balanced weight between the old and new classes, which contradicts the motivation of this paper.

**Suitability:**

2

---

### Official Review · Reviewer_3Mb3 · 2024-05-29

**Rating:** 4
**Confidence:** 2

**Summary:**

The authors propose two methods to improve the loss function, aiming to address class imbalance and the overlap of old and new classes in continual learning.

**Strengths:**

The experimental analysis in this work is thorough, with comprehensive main experimental results and ablation studies.

The motivation for this work is reasonable.

**Limitations:**

I am not familiar with the current state of research in medical image recognition. For a researcher studying imbalanced learning on natural images, the core idea of the class-balanced loss proposed in this work has already been introduced. I am unsure whether the loss function proposed by the authors to address the problem of catastrophic forgetting in continual learning has already been presented by others.

**Suitability:**

2

---

### Official Review · Reviewer_1a1y · 2024-06-09

**Rating:** 4
**Confidence:** 3

**Summary:**

The paper addresses the challenges of class incremental learning in medical image classification, specifically focusing on issues caused by class imbalance. The authors introduce two novel loss functions to mitigate these challenges. The first is a CIL-balanced classification loss that uses logit adjustment to reduce classifier bias towards majority classes. The second is a distribution margin loss designed to enhance inter-class separation and intra-class compactness. Two loss functions can be incorporated in existing training methods in a plug-and-play manner. Experiments on benchmark datasets show that the proposed methods outperform current state-of-the-art approaches.

**Strengths:**

* The paper has a clear motivation and is well organized.
* Proposed loss objectives are very straightforward and easy to implement, and can be incorporated to other prior works in plug-in way.
* Extensive experiments are delivered on three benchmark datasets, and the proposed method outperforms previous state-of-the-art methods with considerable amount of gap.
* Sufficient amount of visualizations and ablation studies are presented to validate each component of the proposed loss objectives.

**Limitations:**

* Overall, it seems like the training methods and pipeline is heavily adopted from the LO2LN [6]. For instance, the combined loss form of margin loss, distillation loss, and class-balanced classification loss is first introduced and validated in the LO2LN. In my perspective, this work proposes a little expanded version of baseline margin loss and class-balanced loss, while maintaining all the rest identical as LO2LN. The added materials on top of baseline margin loss and class-balanced loss are both very straightforward, which are based on category frequency values. However, the performance improvement that stems from them is non-negligible.

* Ablation study on the sample size 'm' of representative instances in memory buffer needs to be explored, since it could be the most important parameter that divides majority and minority class during class incremental training.

**Suitability:**

2

---

### Meta-Review · Area_Chair_EhAw · 2024-07-01

**Recommendation:** Accept (Poster)
**Confidence:** 5

**Metareview:**

This paper proposes a method to address class imbalance in class incremental learning for medical image segmentation. It identifies the imbalance between old and new classes as a major concern and proposes two strategies to mitigate this issue: CIL-balanced classification loss and distribution marginal loss. Experimental results demonstrate the effectiveness of these strategies. This paper will provide valuable insights into the field of class imbalance in class incremental learning.

The authors should incorporate some of the rebuttal content into the revised paper based on the reviewers' suggestions. Based on both the reviewers' comments and the author's rebuttal, I recommend accepting this paper.